# FOCUS ON THE FOG: LEVERAGING STUDENT UNCERTAINTY FOR GUIDED KNOWLEDGE DISTILLATION IN SEMANTIC SEGMENTATION

## ABSTRACT

Current knowledge distillation (KD) methods for semantic segmentation focus on distilling the teacher's knowledge via logit and feature-based techniques. Recent work explored the improvement of knowledge distillation methods by incorporating the uncertainty of the teacher in dense prediction tasks, primarily in object detection. Yet, its application in knowledge distillation for semantic segmentation has received limited attention. Moreover, utilizing the uncertainty on the student side remains largely underexplored. We posit that student-side uncertainty can serve as a valuable signal for guiding the distillation process in semantic segmentation. To this end, we propose Focus on the Fog (FOTF), a novel uncertainty-guided distillation approach that estimates and leverages student-side uncertainty during training. Specifically, we formulate an uncertainty-weighted distillation loss for semantic segmentation that is dynamically modulated by the student's uncertainty, estimated via Monte Carlo Dropout. This amplifies the distillation signal in spatial regions and semantic classes where the student model exhibits low certainty, thereby providing more targeted guidance during training. Extensive experiments on the Cityscapes, CamVid and Pascal VOC datasets demonstrate the effectiveness of our method, both as a standalone technique and as an add-on to existing state-of-the-art knowledge distillation methods. The code will be made publicly available upon acceptance.

## 1 INTRODUCTION

Semantic segmentation is a fundamental task in computer vision with wide-ranging applications, such as autonomous driving Cordts et al. (2016) and medical imaging Ronneberger et al. (2015). Many semantic segmentation models such as DeepLab Chen et al. (2017a; 2018), PSPNet Zhao et al. (2017), and SegFormer Xie et al. (2021) have achieved strong performance across various benchmarks. However, their high computational cost makes them impractical for deployment in real-time or resource-constrained settings, motivating the use of knowledge distillation to transfer performance to lighter models.

Early work on knowledge distillation focused on transferring softened output distributions from a large teacher model to a smaller student model Ba & Caruana (2014); Hinton et al. (2015). By matching these soft targets (Fig. 1a), the student can learn richer inter-class relationships, often referred to as "dark knowledge" Hinton et al. (2015), which are not captured by hard labels. Subsequent work has revisited knowledge distillation from a theoretical perspective, arguing that its effectiveness is not solely due to the similarity information provided by the teacher Yuan et al. (2020). Instead, a significant part of its benefit arises from the regularization effect of the soft targets, framing it as a form of learned label smoothing Yuan et al. (2020). It follows that students can benefit from learning with their own soft targets Yuan et al. (2020).

In semantic segmentation, recent work has increasingly focused on feature-based distillation, with a shift from directly mimicking absolute feature activations to transferring structural dependencies within feature representations Liu et al. (2019); Yang et al. (2022); Fan et al. (2023). This reflects a growing recognition that preserving the relational structure encoded by the teacher is more effective than enforcing strict per-location alignment in dense prediction tasks Liu et al. (2019); Yang et al.

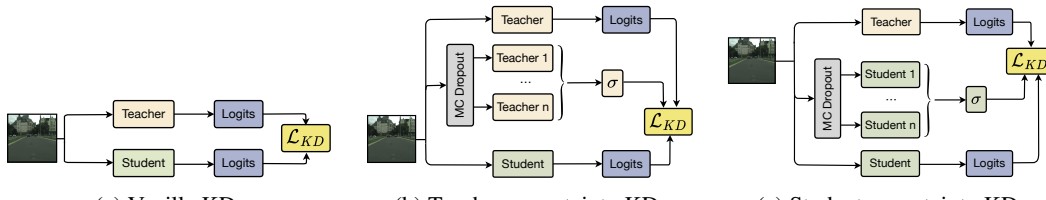

(a) Vanilla KD.    (b) Teacher-uncertainty KD.    (c) Student-uncertainty KD.

Figure 1: Uncertainty-aware knowledge distillation: (a) vanilla; (b) teacher-weighted via MC dropout; (c) our student-weighted via MC dropout.

(2022); Fan et al. (2023). To this end, CIRKD Yang et al. (2022) introduces a framework that aims to preserve better-structured semantic relations both within individual images and across different samples. Af-DCD Fan et al. (2023), in contrast to CIRKD, adopts a buffer-free approach. It employs a masked feature mimicking strategy and performs contrastive learning using both absolute spatial positions and local neighborhoods.

Nevertheless, existing methods largely overlook the uncertainty present in teacher models, which can arise from data noise and imperfect training Yi et al. (2024). One approach addresses this by replacing the teacher with its own inference ensemble, improving the diversity and receptiveness of the knowledge transferred to the student Zhang et al. (2023). Another line of work estimates uncertainty using Monte Carlo Dropout and leverages it to guide student training, encouraging deeper exploration of the latent space Yi et al. (2024). Both of the aforementioned methods estimate uncertainty on the teacher side and use it to guide the student's training (Fig. 1b).

Motivated by the findings of Yuan et al. (2020), which show that students can benefit from learning with their own soft targets, we explore the potential of leveraging uncertainty estimated by the student itself (Fig. 1c). To this end, we introduce Focus on the Fog (FOTF), a novel uncertainty-guided distillation approach that estimates student uncertainty using Monte Carlo (MC) Dropout Gal & Ghahramani (2016) and incorporates it into training via an uncertainty-weighted distillation loss for semantic segmentation (Fig. 1c). We demonstrate that student-driven uncertainty can be a valuable signal during training. This design avoids the overhead of teacher ensembles, offers a more efficient route to distillation, and naturally opens avenues toward active learning. To summarize, we introduce **Focus on the Fog (FOTF), a simple yet effective student-driven uncertainty-guided distillation method** that leverages Monte Carlo Dropout to estimate student uncertainty and integrate it into training; FOTF is **method-agnostic**, works both as a **standalone or as an add-on** to existing distillation techniques, and achieves **consistent improvements on widely used semantic segmentation benchmarks**.

## 2 RELATED WORK

**Knowledge Distillation.** Early work on knowledge distillation focused on transferring softened output distributions from teacher to student models Ba & Caruana (2014); Hinton et al. (2015). Later analyses revealed that much of KD's benefit stems from the regularization effect of soft labels rather than inter-class similarity alone Yuan et al. (2020). Further theoretical investigation frames KD through the lens of the bias–variance trade-off, showing that sample-wise variance can harm distillation and proposing adaptive weighting of soft targets to mitigate this effect Zhou et al. (2021). Beyond classification, KD has been applied to detection and dense prediction. For instance, Li et al. (2017) showed that distilling features from a strong detector can outperform ImageNet pretraining, motivating feature-based transfer. To better capture the structured nature of dense prediction, Liu et al. (2020b) proposed distilling structured knowledge rather than treating pixels independently, achieving stronger generalization. Recent general-purpose methods have explored more flexible or automated strategies. Huang et al. (2022) replaces the standard KL loss with Pearson correlation to better capture relational consistency between student and teacher. Li et al. (2023) introduces KD-Zero, an evolutionary search framework to discover optimal distillation components. Huang et al. (2023a) proposes masking noisy regions in feature maps, focusing distillation on informative spatial areas. DiffKD Huang et al. (2023b) introduces a novel approach where student features are denoised via diffusion models to better match the teacher's representational quality. Multi-teacher distillation has also been explored, with Iordache et al. (2025) proposing the combination of teachers trained on different datasets through a joint fusion module and multi-level feature distillation to improve

generalizability and reduce overfitting.

**Knowledge Distillation in Semantic Segmentation.** Reducing the feature map resolution in semantic segmentation models improves efficiency but leads to notable performance degradation. To address this, He et al. (2019) propose a distillation framework that aligns student and teacher features in a transferred latent space via a pre-trained autoencoder and incorporates an affinity module to capture long-range dependencies. For video segmentation, temporal consistency has been addressed by introducing a temporal loss during training that maintains prediction stability without additional inference cost Liu et al. (2020a). Structured Knowledge Distillation (SKD) Liu et al. (2019) was one of the earliest to tailor feature-based distillation to segmentation, incorporating pairwise losses and a GAN-based holistic loss to preserve spatial and semantic consistency. IFVD Wang et al. (2020) focuses on transferring intra-class variation by aligning pixel-wise features with class-specific prototypes, while CWD Shu et al. (2021) leverages channel-wise importance to guide distillation toward semantically meaningful regions. CIRKD Yang et al. (2022) advances this line of work by enforcing consistency not only within individual images but also across different samples. It aligns pairwise relations in the latent space, encouraging the student to mimic the semantic structure encoded by the teacher. Af-DCD Fan et al. (2023) builds on this idea with a buffer-free contrastive framework. It introduces a masked feature mimicking strategy and formulates a loss across spatial and channel dimensions by leveraging absolute positions and local neighborhoods. Positive pairs are drawn from matching positions, while negative pairs are sampled locally, enabling dense and structured knowledge transfer without memory overhead. Other recent work includes methods that improve teacher quality through noised supervision and dual-path consistency training Qiu et al. (2024), as well as approaches designed for heterogeneous architecture distillation Hao et al. (2023); Huang et al. (2025).

## 3 METHODOLOGIES

**Preliminary.** The vanilla knowledge distillation (Vanilla KD) loss encourages the student network to match the class probabilities predicted by a pre-trained teacher network at each pixel. It is defined as:

$$\mathcal{L}_{\text{KD}} = \frac{1}{H \cdot W} \sum_{i \in \mathcal{R}} \text{KL}(q_i^T \,\|\, q_i^S) \tag{1}$$

$$= \frac{1}{H \cdot W} \sum_{i \in \mathcal{R}} \sum_{c=1}^{C} q_{i,c}^T \cdot \log\left(\frac{q_{i,c}^T}{q_{i,c}^S}\right), \tag{2}$$

where $H$ and $W$ denote the image height and width, respectively, $\mathcal{R} \subseteq \{1, \ldots, H \cdot W\}$ is the set of all pixel locations, and $C$ is the number of semantic classes. For each pixel $i \in \mathcal{R}$, $q_i^T \in [0,1]^C$ and $q_i^S \in [0,1]^C$ represent the softmax output (i.e., class probability distribution) of the teacher and student networks, respectively. The KL divergence is computed at each pixel between the teacher's and student's predicted distributions and averaged over all pixels.

**Uncertainty-Weighted KD Loss.** We propose an extension to the vanilla knowledge distillation loss by incorporating an uncertainty-based weighting scheme. The key idea is to emphasize the learning signal in regions where the student model is uncertain about its predictions. Specifically, for pixels or classes where the student exhibits high uncertainty, we increase their contribution to the distillation loss. Conversely, for confident predictions, the loss contribution remains as in standard KD. In this way, the model is encouraged to focus more on ambiguous regions, while maintaining the original KD behavior for more certain predictions. We formalize this idea through the Uncertainty-Weighted Knowledge Distillation (KD) loss, illustrated in Fig. 2. The modified loss is defined as:

$$\mathcal{L}_{\text{KD}_{\text{unc}}} = \frac{1}{H \cdot W} \sum_{i \in \mathcal{R}} \sum_{c=1}^{C} w(i,c) \cdot q_{i,c}^T \cdot \log\left(\frac{q_{i,c}^T}{q_{i,c}^S}\right), \tag{3}$$

where $w(i,c) \in \mathbb{R}_{\geq 1}$ is an uncertainty-based weighting function defined per pixel $i$ and class $c$. This weight increases the contribution of predictions where the student is more uncertain, encouraging the model to pay closer attention to ambiguous or difficult regions. For certain predictions, the weighting factor approaches unity, thereby reducing to the standard knowledge distillation formulation.

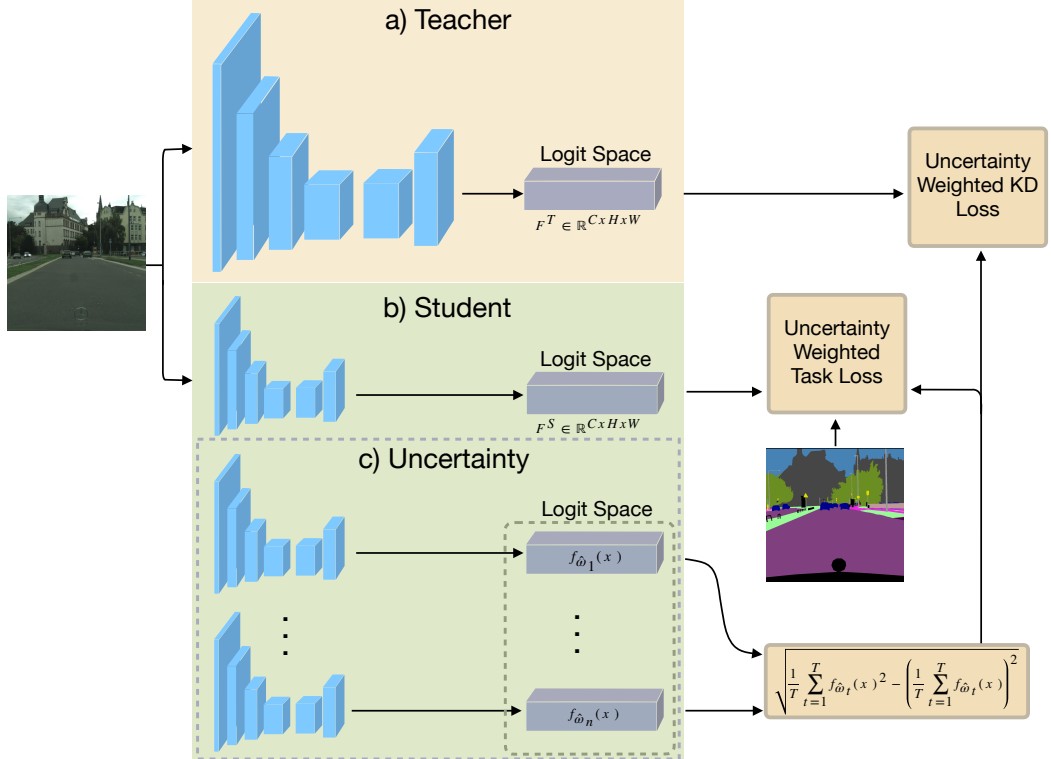

Figure 2: The illustration for the distillation process. To estimate model uncertainty, we apply MC Dropout by performing $T$ stochastic forward passes through the student model with dropout enabled. We then compute the standard deviation of the softmax outputs to quantify uncertainty and apply a monotonically increasing weighting function for proper guidance. This uncertainty is subsequently used to modulate both the Kullback–Leibler divergence loss between the student and teacher ((a) and (c)), as well as the task loss between the student and ground-truth labels ((b) and (c)) with our uncertainty-based weighting function.

**Uncertainty Computation.** To compute the model's uncertainty, we follow a Bayesian approximation approach based on Monte Carlo Dropout as proposed by Gal & Ghahramani (2016); Gal (2016). This method interprets dropout as approximate variational inference in a deep Gaussian Process. The key idea is to obtain an approximation of the model's predictive distribution by performing multiple stochastic forward passes at inference time using dropout. Formally, the predictive distribution is given by:

$$p(y \mid x, \mathcal{D}) \approx \frac{1}{T} \sum_{t=1}^{T} p(y \mid x, \hat{\omega}_t), \tag{4}$$

where $\hat{\omega}_t \sim q(\omega)$ are sampled dropout masks, $q(\omega)$ is a bernoulli-distributed approximate posterior distribution, $p$ refers to the predictive distribution and $T$ is the number of stochastic forward passes.

The corresponding predictive variance, which captures the model's epistemic uncertainty in part, can be approximated as Kendall & Gal (2017):

$$\mathrm{Var}(y) \approx \underbrace{\frac{1}{T} \sum_{t=1}^{T} f_{\hat{\omega}_t}(x)^2 - \left( \frac{1}{T} \sum_{t=1}^{T} f_{\hat{\omega}_t}(x) \right)^2}_{\text{epistemic uncertainty}} + \underbrace{\sigma^2_{aleatoric}}_{\text{aleatoric uncertainty}}, \tag{5}$$

where $f_{\hat{\omega}_t}(x)$ denotes the model's prediction in the $t$-th forward pass.

Since MC Dropout approximates a distribution over model parameters, it captures only epistemic (model) uncertainty. We do not explicitly model aleatoric uncertainty, and therefore treat the predictive variance as a measure of model confidence alone. In practice, we train the segmentation network with dropout enabled Srivastava et al. (2014) and keep dropout active during inference Gal & Ghahramani (2016). We perform $T$ stochastic forward passes and compute the standard deviation of the softmax outputs across these samples, after applying the softmax along the class dimension. This standard deviation is used as a pixel- and channel-wise uncertainty estimate. Formally, for each pixel $i \in \mathcal{R}$ and class $c \in \{1, \ldots, C\}$, we compute the standard deviation of the softmax outputs as:

$$\sigma_{i,c} = \text{std}\left(\left\{q_{i,c}^S\right\}_{t=1}^T\right), \tag{6}$$

where $q_{i,c}^S$ denotes the softmax probability for class $c$ at pixel $i$ in the $t$-th stochastic forward pass. The resulting uncertainty map $u(i, c)$ is used in our loss formulation as an uncertainty-aware weighting factor.

**Weighting Function.** To incorporate uncertainty into the distillation loss in a meaningful way, we define a weighting function where we pass $\sigma_{i,c}$ through a continuous, strictly monotonically increasing transformation $f \colon \mathbb{R}_{\geq 0} \to \mathbb{R}_{\geq 1}$ that satisfies $f(0) = 1$. This ensures that confident predictions ($\sigma_{i,c} \approx 0$) retain their original loss contribution, while more uncertain predictions are up-weighted proportionally. We adopt the following uncertainty-based weighting function, inspired by a formulation previously proposed in the context of uncertainty-guided supervision Stone et al. (2022), and modify it by introducing a scaling factor $m > 0$ to better control the influence of uncertainty in our setting:

$$w(i, c) = (1 + m \cdot \sigma_{i,c})^{\kappa}, \tag{7}$$

where $\sigma_{i,c}$ denotes the standard deviation of the softmax outputs across MC Dropout samples at pixel $i$ and class $c$, and $\kappa > 0$ controls the sharpness of the weighting. We empirically validate the effectiveness of this weighting function in our setting, with ablations provided later in the paper.

**Uncertainty-Weighted Task Loss.** To apply our uncertainty weighting to the supervised task loss, we use the same weighting term $w(i, c)$ to modulate the pixel-wise cross-entropy as follows:

$$
\begin{aligned}
\mathcal{L}_{\text{CE}_{\text{unc}}} &= \frac{1}{H \cdot W} \sum_{i \in \mathcal{R}} \sum_{c=1}^{C} \mathbf{1}_{\{y_i = c\}} \cdot w(i, c) \cdot \left[-\log q_{i,c}^S\right] \\
&= \frac{1}{H \cdot W} \sum_{i \in \mathcal{R}} w(i, y_i) \cdot \left[-\log q_{i,y_i}^S\right],
\end{aligned}
\tag{8}
$$

where $q_{i,y_i}^S$ is the predicted softmax probability of the ground truth class $y_i$ at pixel $i$ and $\mathbf{1}_{\{y_i = c\}}$ denotes the indicator function. This formulation ensures that uncertain predictions are given more emphasis during learning, while confident ones contribute less.

## 4 EXPERIMENTS

### 4.1 IMPLEMENTATION

**Datasets.** Our experiments are conducted on three widely-used semantic segmentation datasets, including Cityscapes Cordts et al. (2016), CamVid Brostow et al. (2008b;a) and Pascal VOC Everingham et al. (2010); Hariharan et al. (2011).

**Network Architectures.** In line with previous work Yang et al. (2022); Fan et al. (2023), we adopt DeepLabV3 Chen et al. (2017a;b), PSPNet Zhao et al. (2017), and SegFormer Xie et al. (2021) as segmentation heads. For the teacher backbone architectures, we use ResNet-101 (Res101) He et al. (2016) and Mix Transformer-B4 (MiT-B4) Xie et al. (2021), while ResNet-18 (Res18) and Mix Transformer-B0 (MiT-B0) are employed as student backbone architectures.

**Evaluation Metrics.** Following the standard setting used in Yang et al. (2022) and Fan et al. (2023), we evaluate segmentation performance using mean Intersection-over-Union (mIoU).

**Distillation Methods.** We compare our approach against state-of-the-art segmentation distillation methods Liu et al. (2019); Wang et al. (2020); Shu et al. (2021); Yang et al. (2022); Fan et al. (2023). To demonstrate the effectiveness of our method as a plug-in component, we integrate it into CIRKD Yang et al. (2022) and Af-DCD Fan et al. (2023).

**Training.** We follow the training and hyperparameter settings used in CIRKD Yang et al. (2022) and

Af-DCD Fan et al. (2023). Unless stated otherwise, we perform $T = 5$ stochastic forward passes using the student model, with a dropout rate of 0.1. To determine the optimal values of $\kappa$ and $m$, we perform a grid-based search over the candidate sets $\kappa \in \{1, .., 10\}$ and $m \in \{1, 3, 10\}$.

## 4.2 RESULTS

**Semantic Segmentation.** Tables 1a, 1b, and 1c present semantic segmentation distillation results on Cityscapes, CamVid, and Pascal VOC using CNN-based students (DeepLabV3-Res18, PSPNet-Res18) with DeepLabV3-Res101 as the teacher. Across all datasets, our proposed Focus on the Fog (FOTF) consistently improves performance when combined with various distillation baselines.

On Cityscapes, FOTF delivers the largest gain: DeepLabV3-Res18 improves from 74.21% mIoU to 77.37% with Vanilla KD + FOTF (+3.16%), the best among all methods. PSPNet-Res18 also benefits, with CIRKD + FOTF and Vanilla KD + FOTF achieving 75.03% and 74.99%, respectively, both surpassing strong baselines. On CamVid, FOTF-enhanced models achieve steady improvements, with DeepLabV3-Res18 reaching 69.66% (+2.74% over baseline)

| Method | Model | Dataset | mIoU ↑ |
|---|---|---|---|
| Baseline | | Cityscapes | 74.21 |
| + FOTF | | Cityscapes | **75.07** (+0.86) |
| Baseline | DeepLabV3-Res18 | CamVid | 66.92 |
| + FOTF | | CamVid | **67.91** (+0.99) |
| Baseline | | PascalVOC | 73.21 |
| + FOTF | | PascalVOC | **74.27** (+1.06) |
| Baseline | | Cityscapes | 72.55 |
| + FOTF | | Cityscapes | **72.85** (+0.30) |
| Baseline | PSPNet-Res18 | CamVid | 66.73 |
| + FOTF | | CamVid | **67.91** (+1.18) |
| Baseline | | PascalVOC | 73.33 |
| + FOTF | | PascalVOC | **73.95** (+0.62) |

Table 2: Semantic segmentation results across all datasets for the baseline and baseline with uncertainty (without knowledge distillation, only equation 8).

using Vanilla KD + FOTF. For PSPNet-Res18, Af-DCD + FOTF achieves the best mIoU of 69.63%, while other FOTF variants remain competitive.

On Pascal VOC, where the base DeepLabV3-Res18 model achieves 73.21% mIoU, Af-DCD remains the strongest performer with 76.25%, followed by CIRKD + FOTF at 74.70%. Although Vanilla KD + FOTF only reaches 74.21%, it still outperforms all other baseline methods except Af-DCD and CIRKD, demonstrating the benefit of integrating FOTF even with a simple distillation setup. For PSPNet-Res18, Af-DCD again achieves the highest score (76.14%), followed by CIRKD + FOTF (74.97%). Adding FOTF to CIRKD yields consistent improvements, with an average gain of +0.20% mIoU across both architectures.

Table 2 shows the effect of applying FOTF directly to the

| Method | Params ↓ | mIoU ↑ |
|---|---|---|
| T: SegFormer-MiT-B4 | 64.1M | 81.23 |
| S: SegFormer-MiT-B0 | | 75.58 |
| + SKD | | 76.43 (+0.85) |
| + IFVD | 3.8M | 76.30 (+0.72) |
| + CWD | | 74.80 (-0.78) |
| + CIRKD | | **76.92** (+1.34) |
| + Af-DCD | | 75.89 (+0.31) |
| S: SegFormer-MiT-B0 | | 75.93 |
| + FOTF | 3.8M | **76.71** (+0.78) |
| + CIRKD | | 75.93 (+0.00) |
| + CIRKD + FOTF | | 76.31 (+0.38) |

Table 3: Semantic segmentation distillation results on the Cityscapes validation set using SegFormer-MiT-B0 (student) with SegFormer-MiT-B4 (teacher). The first subtable presents the baselines reported in the literature, while the second shows our reproduced baselines with dropout and our proposed add-on extensions. Params from Xie et al. (2021). **Bold** = best, underline = second-best.

baseline models without any additional distillation. The best improvement is observed with PSPNet-Res18 on CamVid, achieving a gain of +1.18% mIoU, while the smallest improvement occurs on Cityscapes with the same architecture (+0.30% mIoU). On average, applying FOTF yields a con-

| Method | Params ↓ | mIoU ↑ |
|---|---|---|
| T:DeepLabV3-Res101 | 61.1M | 78.07 |
| S:DeepLabV3-Res18 | | 74.21 |
| +SKD | | 75.42 (+1.21) |
| +IFVD | 13.6M | 75.59 (+1.38) |
| +CWD | | 75.55 (+1.34) |
| +CIRKD | | 76.38 (+2.17) |
| +Af-DCD | | 77.03 (+2.82) |
| +Vanilla KD +FOTF | | **77.37** (+3.16) |
| +CIRKD +FOTF | 13.6M | 76.43 (+2.22) |
| +Af-DCD +FOTF | | 76.93 (+2.72) |
| S:PSPNet-Res18 | | 72.55 |
| +SKD | | 73.29 (+0.74) |
| +IFVD | 12.9M | 73.71 (+1.16) |
| +CWD | | 74.36 (+1.81) |
| +CIRKD | | 74.73 (+2.18) |
| +Af-DCD* | | 74.21 (+1.66) |
| +Vanilla KD +FOTF | | 74.99 (+2.44) |
| +CIRKD +FOTF | 12.9M | **75.03** (+2.48) |
| +Af-DCD +FOTF | | 74.75 (+2.20) |

(a) Cityscapes

| Method | Params ↓ | mIoU ↑ |
|---|---|---|
| T:DeepLabV3-Res101 | 61.1M | 69.84 |
| S:DeepLabV3-Res18 | | 66.92 |
| +SKD | | 67.46 (+0.54) |
| +IFVD | 13.6M | 67.28 (+0.36) |
| +CWD | | 67.71 (+0.79) |
| +CIRKD | | 68.21 (+1.29) |
| +Af-DCD | | 69.27 (+2.35) |
| +Vanilla KD +FOTF | | **69.66** (+2.74) |
| +CIRKD +FOTF | 13.6M | 69.33 (+2.41) |
| +Af-DCD +FOTF | | 69.63 (+2.71) |
| S:PSPNet-Res18 | | 66.73 |
| +SKD | | 67.83 (+1.10) |
| +IFVD | 12.9M | 67.61 (+0.88) |
| +CWD | | 67.92 (+1.19) |
| +CIRKD | | 68.65 (+1.92) |
| +Af-DCD | | 69.48 (+2.75) |
| +Vanilla KD +FOTF | | 68.93 (+2.20) |
| +CIRKD +FOTF | 12.9M | 69.08 (+2.35) |
| +Af-DCD +FOTF | | **69.63** (+2.90) |

(b) CamVid

| Method | Params ↓ | mIoU ↑ |
|---|---|---|
| T:DeepLabV3-Res101 | 61.1M | 77.67 |
| S:DeepLabV3-Res18 | | 73.21 |
| +SKD | | 73.51 (+0.30) |
| +IFVD | 13.6M | 73.85 (+0.64) |
| +CWD | | 74.02 (+0.81) |
| +CIRKD | | 74.50 (+1.29) |
| +Af-DCD | | **76.25** (+3.04) |
| +Vanilla KD +FOTF | 13.6M | 74.21 (+1.00) |
| +CIRKD +FOTF | | 74.70 (+1.49) |
| S:PSPNet-Res18 | | 73.33 |
| +SKD | | 74.07 (+0.74) |
| +IFVD | 12.9M | 73.54 (+0.21) |
| +CWD | | 73.99 (+0.66) |
| +CIRKD | | 74.78 (+1.45) |
| +Af-DCD | | **76.14** (+2.81) |
| +Vanilla KD +FOTF | 12.9M | 74.22 (+0.89) |
| +CIRKD +FOTF | | 74.97 (+1.64) |

(c) Pascal VOC

Table 1: Semantic segmentation distillation results on Cityscapes, CamVid, and Pascal VOC using DeepLabV3-Res18 and PSPNet-Res18 as students with DeepLabV3-Res101 as teacher. Vanilla KD uses KL divergence. Params from Yang et al. (2022). Best in **bold**, second-best underlined. *Reproduced using the official code.

sistent boost of +0.84% mIoU across datasets and architectures, demonstrating its standalone effectiveness even without teacher supervision.

Furthermore, Table 3 presents results for a high-performing transformer-based segmentation architecture—SegFormer-MiT-B0 as the student and MiT-B4 as the teacher—on the Cityscapes val-

| $\kappa$ merge | logits | | pixel | | channel | | sample | |
|---|---|---|---|---|---|---|---|---|
| | $(1+\sigma)^{\kappa}$ | $e^{\kappa\sigma}$ | $(1+\sigma)^{\kappa}$ | $e^{\kappa\sigma}$ | $(1+\sigma)^{\kappa}$ | $e^{\kappa\sigma}$ | $(1+\sigma)^{\kappa}$ | $e^{\kappa\sigma}$ |
| 1 | 75.52 | 75.93 | 75.69 | 76.28 | 75.83 | 75.78 | 75.40 | 75.33 |
| 3 | 76.14 | 75.96 | 75.95 | 75.96 | 76.46 | 75.95 | 76.09 | 75.54 |
| 5 | 75.58 | 75.61 | 75.64 | 76.09 | 76.13 | 75.72 | 76.06 | 76.00 |
| average | 75.75 | 75.83 | 75.76 | 76.11 | **76.14** | 75.82 | 75.85 | 75.62 |

Table 4: Semantic segmentation performance on Cityscapes for DeepLabV3 using different uncertainty merging strategies across various aggregation dimensions: logits, pixels, channels, and samples. For each dimension, we compare two weighting functions—$(1+\sigma)^{\kappa}$ and $e^{\kappa\sigma}$—and evaluate results for $\kappa \in \{1,3,5\}$. As a reference, the vanilla knowledge distillation baseline yields a score of 75.65 % mIoU. The best-performing method is shown in bold, and the second-best is underlined.

idation set. The uncertainty-guided approach (FOTF) improves both the baseline model and its combination with CIRKD. Specifically, applying FOTF alone boosts the baseline from 75.93% to 76.71%, while CIRKD + FOTF achieves 76.31%, outperforming the CIRKD baseline reproduced with dropout. Overall, the proposed method remains highly competitive, with the exception of the case where CIRKD without dropout slightly outperforms the FOTF-enhanced baseline (76.92%).

**Ablation Analysis.** The results for ablation experiments with different uncertainty merging strategies and weighting functions are presented in Table 4. Here, merging denotes the process of averaging uncertainty estimates across a particular dimension. Nearly all configurations that incorporate uncertainty outperform vanilla knowledge distillation (75.65% mIoU), demonstrating the consistent benefit of uncertainty guidance. The two best-performing configurations are channel-wise merging with $(1+\sigma)^{\kappa}$ (76.14% mIoU) and pixel-wise merging with $e^{\kappa\sigma}$ (76.11% mIoU), with the former selected as the default due to its superior performance.

Building upon this, Table 5 investigates whether applying uncertainty to the task loss provides additional gains. Using the best configuration from the previous ablation as a base (76.14% mIoU), adding pixel-wise uncertainty-weighted task loss yields an improved score of 76.28% mIoU. These results confirm that once uncertainty is incorporated into the distillation objective, extending it to the task loss is beneficial.

| $\kappa$ uncert. | yes | no |
|---|---|---|
| 1 | 76.08 | 75.83 |
| 3 | 76.30 | 76.46 |
| 5 | 76.45 | 76.13 |
| average | **76.28** | 76.14 |

Table 5: Ablation study evaluating the impact of applying uncertainty-weighted task loss, using pixel-wise uncertainty merging and the weighting function $(1+\sigma)^{\kappa}$. Best in bold.

Table 6 presents the additional relative training time when performing MC Dropout with varying numbers of stochastic forward passes on the teacher and student models. As expected, the training time increases significantly when MC Dropout is applied to both sides. Using 5 stochastic passes on the student increases runtime to 152.9 (152.9 relative to the baseline of 100), while applying the same to

| student teacher | 0 | 5 | 10 | 15 | 20 |
|---|---|---|---|---|---|
| 0 | 100.0 | 152.9 | 205.7 | 260.0 | 312.8 |
| 5 | 342.9 | 395.7 | 450.0 | 521.4 | 564.3 |

Table 6: Profiling results (relative runtime; 100 = reference) for Vanilla KD and MC Dropout on Cityscapes with varying numbers of stochastic forward passes for both teacher and student. Measured over 50 iterations on an RTX 3090 Ti.

the teacher leads to a significantly higher cost of 342.9. Even with 20 passes, student-only dropout remains more efficient at 312.8—still below the cost of applying just 5 passes to the teacher. The approach adopted in this work relies solely on student-side uncertainty, which provides a favorable trade-off between performance and efficiency. This design avoids additional computational burden on the teacher and ensures that the distillation process remains scalable, while still benefiting from

uncertainty-guided learning.

**Expected Calibration Error.** Table 7 reports the expected calibration error (ECE) Guo et al. (2017) alongside mIoU for various methods, with and without the proposed FOTF enhancement.

In all cases, FOTF leads to a reduction in ECE, indicating improved calibration. However, the improvements are relatively small (e.g., –0.14 for the baseline, –0.17 for CIRKD, and –0.02 for Af-DCD), and therefore not significant enough to draw strong conclusions. The slight decrease in ECE observed in some of the high-performing models may be attributed to the increase in overall accuracy. Nonetheless, a key takeaway is that FOTF does not degrade model calibration, and in most cases, results in modest improvements.

| Method | mIOU ↑ | ECE ↓ |
|--------|--------|-------|
| Baseline | 66.96 | 4.46 |
| +FOTF | **67.91** | **4.32** (-0.14) |
| CIRKD | 68.15 | 3.51 |
| +FOTF | **69.33** | **3.34** (-0.17) |
| Af-DCD | 69.27 | 3.28 |
| +FOTF | **69.63** | **3.26** (-0.02) |

Table 7: Semantic segmentation performance (mIoU) and expected calibration error (ECE) Guo et al. (2017) for the baseline and its variants augmented with our proposed FOTF method, across all datasets. ECE is computed over the full CamVid test set using 10 bins.

## 5 LIMITATIONS

MC Dropout adds training overhead due to multiple stochastic passes, but this remains manageable as it is applied only on the student and can be further reduced with efficient implementation. MC Dropout may also lack reliability in estimating uncertainty, where alternative methods could offer more robust estimates and warrant further investigation. In some cases, such as DeepLabV3 with Af-DCD on Cityscapes, the proposed method does not produce an improvement (e.g. –0.1% mIoU), indicating that uncertainty-guided weighting may not always enhance performance. However, the overall results suggest that this direction remains promising and merits continued exploration. Lastly, the current use of grid-based search for hyperparameters such as $\kappa$ and $m$ may not be optimal and adaptive or data-driven parameter selection techniques could further improve performance.

## 6 CONCLUSION

In this paper, we propose Focus on the Fog (FOTF), a novel uncertainty-guided distillation approach that incorporates the student uncertainty into the distillation training. Comprehensive experiments on both CNN-based and transformer-based architectures demonstrate the effectiveness of our method in improving student model performance across a variety of datasets and distillation frameworks. Albeit the identified limitations, relying solely on student-side uncertainty emerges as a viable and efficient training signal. Our approach is model-agnostic and can be easily applied to existing KD methods, holding promise for broader applicability beyond semantic segmentation. Future work could extend this strategy to other tasks and architectures, while also addressing current limitations such as more reliable uncertainty estimation and adaptive weighting schemes.

## LLM USAGE

We used a large language model (LLM) to assist in polishing the writing and improving readability of the manuscript, as well as in formatting tables for clarity.

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

## A  APPENDIX

### A.1  DATASETS

**Cityscapes.** Cordts et al. (2016) is an urban scene parsing dataset that contains 5000 finely annotated images, where 2975/500/1525 images are used for train/val/test. The segmentation performance is reported on 19 classes.

**CamVid.** Brostow et al. (2008b;a) is an automotive dataset that contains 367/101/233 images for train/val/test with 11 semantic classes.

**Pascal VOC.** Everingham et al. (2010) is a visual object segmentation dataset, which contains 20 foreground classes and 1 background class. Following Yang et al. (2022); Fan et al. (2023), we employ the augmented dataset with extra annotations provided by Hariharan et al. (2011) resulting in 10582/1449 images for train/val.

### A.2  TRAINING DETAILS

**DeeplabV3, PSPNet.** We follow the general settings in Yang et al. (2022); Fan et al. (2023). Random flipping and scaling in the range of [0.5, 2] are employed to augment the data. All experiments are optimized by SGD with a momentum of 0.9, a batch size of 16, an initial learning rate of 0.02 and a weight decay of 0.0001. The number of total training iterations is 40K. The learning rate is decayed by $(1 - \frac{iter}{total\_iter})^{0.9}$ following the polynomial annealing policy Chen et al. (2017b). For crop size during the training phase, we use 512×1024, 360×360 and 512×512 for Cityscapes, CamVid and Pascal VOC, respectively.

**SegFormer.** We follow the general settings in Yang et al. (2022). Random flipping and scaling in the range of [0.5, 2] are employed to augment the data. All experiments are optimized by AdamW **??** with a batch size of 8, an initial learning rate of 0.0002 and a weight decay of 0.0001. The number of total training iterations is 160K. The learning rate is decayed by $(1 - \frac{iter}{total\_iter})^{0.9}$ following the polynomial annealing policy Chen et al. (2017b). For crop size during the training phase, we use 1024×1024 for Cityscapes.

## A.3 HYPERPARAMETER SETTINGS

### A.3.1 CITYSCAPES

**DeepLabV3.** We employ the following $(\kappa, m)$ tuples: $(5, 1)$ for Baseline+FOTF, $(5, 1)$ for Vanilla KD+FOTF, $(1, 1)$ for CIRKD+FOTF and $(7, 1)$ for Af-DCD+FOTF. Moreover, for Vanilla KD + FOTF, we increase the weight decay to 0.0005.
**PSPNet.** We employ the following $(\kappa, m)$ tuples: $(7, 1)$ for Baseline+FOTF, $(6, 1)$ for Vanilla KD+FOTF, $(5, 1)$ for CIRKD+FOTF and $(5, 1)$ for Af-DCD+FOTF. Moreover, for Vanilla KD + FOTF, we increase the weight decay to 0.0005.
**SegFormer.** We employ the following $(\kappa, m)$ tuples: $(5, 1)$ for Baseline+FOTF and $(1, 1)$ for CIRKD+FOTF.

### A.3.2 CAMVID

**DeepLabV3.** We employ the following $(\kappa, m)$ tuples: $(3, 1)$ for Baseline+FOTF, $(10, 10)$ for Vanilla KD+FOTF, $(7, 1)$ for CIRKD+FOTF and $(1, 3)$ for Af-DCD+FOTF.
**PSPNet.** We employ the following $(\kappa, m)$ tuples: $(1, 1)$ for Baseline+FOTF, $(2, 1)$ for Vanilla KD+FOTF, $(10, 1)$ for CIRKD+FOTF and $(3, 3)$ for Af-DCD+FOTF.

### A.3.3 PASCAL VOC

**DeepLabV3.** We employ the following $(\kappa, m)$ tuples: $(9, 1)$ for Baseline+FOTF, $(3, 1)$ for Vanilla KD+FOTF, $(3, 1)$ for CIRKD+FOTF. Moreover, we decrease the learning rate to 0.015 for CIRKD+FOTF. For Af-DCD, we were unable to reproduce the reported results due to high variance in training.
**PSPNet.** We employ the following $(\kappa, m)$ tuples: $(3, 3)$ for Baseline+FOTF, $(3, 1)$ for Vanilla KD+FOTF, $(1, 1)$ for CIRKD+FOTF. Moreover, we decrease the learning rate to 0.015 for Vanilla KD+FOTF and to 0.016 for CIRKD+FOTF. For Af-DCD, we were unable to reproduce the reported results due to high variance in training.

## A.4 ABLATION: UNCERTAINTY MERGING FOR TASK LOSS

Table 8 presents the ablation results for different uncertainty merging strategies in the task loss. On average, the pixel-wise variant achieves the highest score across merging strategies. In comparison, the sample-wise approach performs reasonably well but remains inferior to pixel-wise, while the channel-wise variant fails to produce competitive results.

| $\kappa$ merge | pixel | channel | sample |
|---|---|---|---|
| 1 | 76.08 | 75.30 | 75.79 |
| 3 | 76.30 | 74.99 | 76.44 |
| 5 | 76.45 | 74.59 | 75.27 |
| average | **76.28** | 74.96 | 75.83 |

Table 8: Ablation study on uncertainty merging strategies for task loss, comparing pixel-wise, channel-wise, and sample-wise variants across different values of $\kappa$. The best-performing method is shown in bold, and the second-best is underlined.

## A.5 T-SNE VISUALIZATION

Figure 3, shows a t-SNE visualization of the learned feature embeddings on the Cityscapes dataset for CIRKD (Fig. 3a) and CIRKD+FOTF (Fig. 3b). In the dense central region, where classes overlap and higher uncertainty would be expected, the uncertainty-based CIRKD+FOTF produces tighter clusters, indicating improved separation of challenging, ambiguous samples.

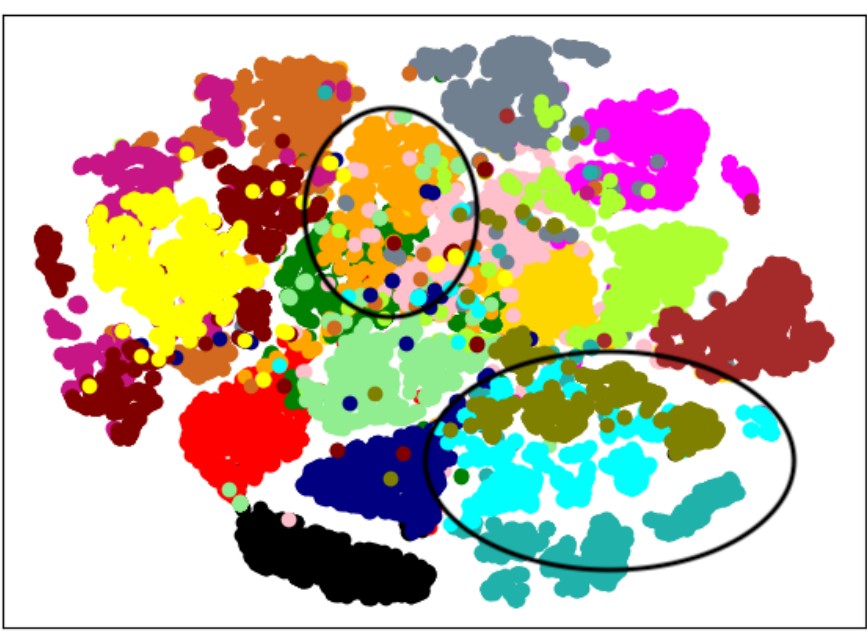

(a) CIRKD.

(b) CIRKD + FOTF.

Figure 3: T-SNE visualization of learned feature embeddings on Cityscapes Cordts et al. (2016) using PSPNet Zhao et al. (2017).

