# OpenReview forum: "Focus on the Fog: Leveraging Student Uncertainty for Guided Knowledge Distillation in Semantic Segmentation"
_ICLR.cc/2026/Conference — ICLR 2026 Conference Withdrawn Submission_

### Official Review · Reviewer_dPx4 · 2025-10-31

**Soundness:** 3
**Presentation:** 2
**Contribution:** 2
**Rating:** 2
**Confidence:** 5

**Summary:**

This paper explores the use of student prediction uncertainty in knowledge distillation of semantic segmentation. While the paper is well-written and presents extensive experiments, its core contributions are weakened by a lack of clear motivation, limited performance gains over SOTA, and unconvincing claims. Significant formatting issues also detract from the paper's readability.

**Strengths:**

1. The paper is well-written and easy to follow. The methodology and experiments setups are presented in a clear manner.

2. The work investigates a potentially interesting direction by exploring the use of uncertainty derived from the student network's predictions for distillation, rather than relying solely on the teacher.

3. The authors have conducted extensive and thorough experiments to evaluate their proposed method across multiple datasets and benchmarks.

**Weaknesses:**

1. Weak Motivation: The primary weakness of this paper is its motivation. The authors do not provide a clear justification, either through theoretical analysis or preliminary experiments, for why student uncertainty should be a valuable signal for distillation compare to utilizing  teacher uncertainty. Furthermore, the paper lacks an in-depth analysis of the mechanism by which this form of distillation works, leaving the reader to question why it is effective (if at all).

2. Chaotic Formatting and Layout: The paper's presentation is severely hampered by formatting issues, which appear to deviate from the ICLR template. The table ordering is confusing: Table 1 is placed after Tables 2, 3.
Tables 2, 3,4 and 5 seem to use a floating layout, which breaks the flow of the results section and makes it extremely difficult to read and connect the text to the corresponding results. This modification of the standard template significantly harms readability.

3. Limited Performance Improvement: The empirical results, while extensive, show only marginal improvements. Compared to current state-of-the-art (SOTA) distillation methods, the performance gains are minimal <0.5 mIOU. The method does not appear to function as a "plugin" that can be added to other methods for additional performance benefits, limiting its practical utility.

4. Unconvincing Research Claims: The three main contributions claimed by the paper are not well-supported:
Claim 1 (Uncertainty Sampling): The logic for uncertainty sampling seems to be identical to or highly similar to methods based on teacher uncertainty.
Claim 2 (Method-Agnostic): The claim of the method being "method-agnostic" is a significant overstatement (miss-claim). When applied to a strong baseline like af-DCD, the method yields negligible improvements of only 0.34 on Cityscapes and 0.39 on CamVid, suggesting it does not generalize well as an add-on.
Claim 3 (Performance): As stated in point 3, the experimental performance gains over SOTA are too limited to be considered a significant contribution.

**Questions:**

1. Could you provide a more detailed explanation (and preferably, experimental support) for the core motivation? Specifically, why is student uncertainty a more beneficial signal for distillation than other existing signals? What is the underlying mechanism that makes this transfer effective?

2. The paper introduces student uncertainty, but methods using teacher uncertainty already exist. Could you clarify the specific scenarios or conditions under which using student uncertainty (as proposed) would be preferable to using teacher uncertainty?

3. The results show very limited gains (e.g., ~0.3-0.4) when this method is combined with af-DCD. Given this, could you better define the conditions under which your method is intended to be used as a "plug-in"? When should a researcher choose to use this method as an add-on, versus simply using a different, stronger baseline method?

---

### Official Review · Reviewer_bX6u · 2025-10-31

**Soundness:** 1
**Presentation:** 2
**Contribution:** 1
**Rating:** 2
**Confidence:** 4

**Summary:**

This work address the insufficient exploration of the potential of student-model uncertainty in knowledge distillation. It estimates the student's uncertainty using Monte Carlo Dropout and dynamically adjusts the weights of the distillation loss and task loss accordingly. This allows the model to receive stronger learning signals in regions and categories where predictions are uncertain, marking the first systematic introduction of student-side uncertainty into semantic segmentation distillation.

**Strengths:**

1) The paper has a clear structure, with well-presented figures and formulas.

2) The method description and experimental setup are detailed and easy to understand.

**Weaknesses:**

1）The approach is still built upon existing uncertainty estimation (MC Dropout) and weighted loss frameworks, which have been extensively explored in prior works, lacking sufficient novelty [Ref.1], [Ref.2], [Ref.3].

2）The performance improvement is limited and unstable. As shown in Table 2, the gains are small, with most experimental results showing less than a 1% improvement.

3）The proposed uncertainty-based method indirectly requires storing multiple student models, leading to high training overhead.

[Ref.1]Ling Ge, Chunming Hu, Guanghui Ma, Jihong Liu, Hong Zhang,Uncertainty-Aware Multi-Shot Knowledge Distillation for Image-Based Object Re-Identification,AAAI 2020:18056-18064.
[Ref.2] Xin Jin, Cuiling Lan, Wenjun Zeng, Zhibo Chen.Discrepancy and Uncertainty Aware Denoising Knowledge Distillation for Zero-Shot Cross-Lingual Named Entity Recognition.AAAI 2024: 11165-11172.
[Ref.3]Luyang Fang, Yongkai Chen, Wenxuan Zhong.Bayesian Knowledge Distillation: A Bayesian Perspective of Distillation with Uncertainty Quantification. ICML 2024.

**Questions:**

1）In which scenarios or categories does the student model exhibit the highest uncertainty? Does it correspond to “ambiguous regions” in the image (e.g., edges, small objects, or occlusions)? Please provide additional visualization analysis.

2）Have you considered using uncertainty estimation methods other than MC Dropout (such as Deep Ensembles or Bayesian Neural Networks)? Have you compared their effectiveness and efficiency?

3）The paper lacks comparisons with recent distillation methods that also focus on uncertainty [Ref.1], [Ref.2], [Ref.3]

---

### Official Review · Reviewer_DTKq · 2025-10-31

**Soundness:** 3
**Presentation:** 2
**Contribution:** 2
**Rating:** 2
**Confidence:** 3

**Summary:**

This paper introduces a student-uncertainty-guided approach to knowledge distillation in semantic segmentation. The key contribution is the use of uncertainty estimated from the student model as a weighting signal to modulate both the distillation loss and the supervised task loss. Thus, focusing the student’s learning on spatial regions and classes with higher uncertainty. The experiments are performed on Cityscapes, CamVid, and Pascal VOC using both CNN and transformer segmentation architectures. Ablation studies, calibration analysis, and implementation details are provided.

**Strengths:**

1. Clear motivation: The core idea of modulating distillation and supervised losses using student-side uncertainty is clearly articulated, mathematically formulated, and positioned as an efficient alternative to teacher-uncertainty-based approaches.
2. Thorough empirical evaluation: The method is benchmarked on strong datasets (Cityscapes, CamVid, Pascal VOC) using both CNN and transformer architectures, yielding consistent, non-trivial improvements across several baseline and SOTA distillation frameworks

**Weaknesses:**

1. The primary conceptual novelty by weighting losses of student-side uncertainty is somewhat incremental relative to established practices in uncertainty-aware learning and teacher-uncertainty distillation. There is an absence of a sharp theoretical justification or deeper analysis; the method is empirically driven rather than theoretically grounded.
2. Some recent, directly relevant papers on multi-source uncertainty, focal knowledge distillation, and domain-adaptive segmentation are not discussed.
3. The manuscript's layout could be improved by optimizing the presentation of figures and tables for space efficiency. For example, Figure 2 and Table 1(c) are currently formatted in a way that consumes excessive page space and could be redesigned for a more compact and reader-friendly layout.

**Questions:**

1. Can the authors provide deeper quantitative or qualitative insights into where and why student-side uncertainty weighting yields improvements? For example, are there specific classes, spatial contexts, or error modes for which FOTF is particularly beneficial or detrimental?
2. Would alternative uncertainty estimation methods (e.g., ensembling, test-time augmentation, or evidential deep learning) outperform MC Dropout in this context, or provide complementary signals? Any initial experiments or intuition?
3. The design of the weighting function $w(i, c)$ is empirical. Have the authors considered data-driven or adaptive schemes (e.g., learning the weighting) rather than fixed functional forms?
4. Is there a risk of overfitting or error magnification in sparse/ambiguous semantic regions when up-weighting uncertain predictions? How stable are results to variation in $m$, $\kappa$, or number of MC passes?

---

### Note · Authors · 2025-11-17

I have read and agree with the venue's withdrawal policy on behalf of myself and my co-authors.